

# Boxer crabs induce asexual reproduction of their associated sea anemones by splitting and intraspecific theft

Yisrael Schnytzer[1,*], Yaniv Giman[1,*], Ilan Karplus[2] and Yair Achituv[1]

[1] The Mina & Everard Goodman Faculty of Life Sciences, Bar-Ilan University, Ramat-Gan, Israel
[2] Institute of Animal Science, Agricultural Research Organization, The Volcani Center, Rishon Lezion, Israel
[*] These authors contributed equally to this work.

Corresponding author
Yisrael Schnytzer,
newsroolchy@gmail.com

## ABSTRACT

Crabs of the genus *Lybia* have the remarkable habit of holding a sea anemone in each of their claws. This partnership appears to be obligate, at least on the part of the crab. The present study focuses on *Lybia leptochelis* from the Red Sea holding anemones of the genus *Alicia* (family Aliciidae). These anemones have not been found free living, only in association with *L. leptochelis*. In an attempt to understand how the crabs acquire them, we conducted a series of behavioral experiments and molecular analyses. Laboratory observations showed that the removal of one anemone from a crab induces a "splitting" behavior, whereby the crab tears the remaining anemone into two similar parts, resulting in a complete anemone in each claw after regeneration. Furthermore, when two crabs, one holding anemones and one lacking them, are confronted, the crabs fight, almost always leading to the "theft" of a complete anemone or anemone fragment by the crab without them. Following this, crabs "split" their lone anemone into two. Individuals of *Alicia* sp. removed from freshly collected *L. leptochelis* were used for DNA analysis. By employing AFLP (Fluorescence Amplified Fragments Length Polymorphism) it was shown that each pair of anemones from a given crab is genetically identical. Furthermore, there is genetic identity between most pairs of anemone held by different crabs, with the others showing slight genetic differences. This is a unique case in which one animal induces asexual reproduction of another, consequently also affecting its genetic diversity.

Subjects Animal Behavior, Ecology, Genetics, Marine Biology, Zoology
Keywords Boxer crabs, Asexual reproduction, AFLP, Symbiosis, Sea anemone, Lybia

## INTRODUCTION

Boxer crabs of the genus *Lybia* have the remarkable habit of carrying a sea anemone in each of its claws by means of delicate hooks, slightly embedded in the sea anemone column (*Duerden, 1905*; *Guinot, 1976*; *Schnytzer et al., 2013*). *Lybia* gain both nutritional and protective benefits from their sea anemones (*Duerden, 1905*; *Karplus, Fiedler & Ramcharan, 1998*; *Schnytzer et al., 2013*). Although crab-cnidarian associations are generally characterized by a small crab and a larger cnidarian associate who is regarded as the clear host (*Thiel & Baeza, 2001*), in this case, the crab is the larger of the two associates, making the host-symbiont identification harder to define. Due to this "inverted" situation,

the crab which is the larger of the two associates effectively controls the movement of its "host" sea anemone. Previous studies have often suggested that the crab-held sea anemones gain in addition to mobility, transport to further food sources and oxygen (*Duerden, 1905*; *Karplus, Fiedler & Ramcharan, 1998*; *Schnytzer et al., 2013*). However, in a previous study, we showed that the crabs regulate the food intake of their sea anemones, and consequently control their growth, maintaining small, "bonsai" sea anemones for their use (*Schnytzer et al., 2013*).

The association between boxer crabs and sea anemones occurs in two genera, *Lybia* and *Polydectus*, both members of the subfamily Polydectinae Dana, 1852 (*Guinot, 1976*; *Guinot, Doumenc & Chintiroglou, 1995*; *Chen & Hsueh, 2007*). We studied *Lybia leptochelis* from the Red Sea. The sea anemones held by *L. leptochelis* have been identified as an unrecognized *Alicia* that has not been found freely living (DG Fautin & AL Crowther, pers. comm., 2008). The partnership between *L. leptochelis* and *Alicia* sp. appears to be obligate, at least on part of the crab, as we have never observed a crab in nature without sea anemones (*n* > 100), including juvenile crabs not long after settling from their planktonic larval stage (*Schnytzer et al., 2013*). In contrast to *L. leptochelis*, the sea anemone that is mostly associated with *Lybia* crabs is *Triactis producta* (*Duerden, 1905*; *Cutress, 1977*; *Karplus, Fiedler & Ramcharan, 1998*). *T. producta* is widely distributed in tropical seas, and in the Red Sea it is found growing on the base of branching corals in shallow waters (*Fishelson, 1970*; Y Schnytzer, pers. obs., 2010). Most *Lybia* inhabit the upper infralittoral zone in and around coral reefs, with access to *T. producta*, including *L. leptochelis.* However, in the Red Sea they are only found holding *Alicia* sp. (*Schnytzer et al., 2013*). When deprived of their sea anemones, the crabs make no use of their delicate claws but use their first walking legs, and sometimes the second and third ones, for the gathering of food and other behaviors usually performed by the claws (*Duerden, 1905*; *Karplus, Fiedler & Ramcharan, 1998*; *Schnytzer et al., 2013*). Crabs held in the laboratory without sea anemones, but provided with *ad libitum* food are able to survive for several months (*Schnytzer et al., 2013*). However, due to their "sea anemone holding" adapted claws, their inability to gather food and defend themselves in typical crab fashion, makes them unlikely to survive for long in the wild without the sea anemones.

Sea anemones are diverse and successful anthozoans, found in all marine habitats and at all depths and latitudes. Their ecological success is undoubtedly enhanced by their propensity for engaging in symbiotic relationships with other animals, such as unicellular photosynthetic algae, hermit crabs, mollusks, and clown fish (*Daly et al., 2008*). The life cycles of many sea anemones regularly feature, along with sexual reproduction, some form of asexual propagation (reviews by *Chia, 1976*; *Shick, 1991*). The occurrence and mode of asexual propagation, whether via budding, fission, pedal laceration, or apomictic parthenogenesis, varies among families, genera, and even sister-species within the same genus (*Chia, 1976*; *Francis, 1988*; *Shick, 1991*), suggesting that asexual multiplication has a complex evolutionary history among sea anemones (*McFadden et al., 1997*). Like many facultative asexual organisms (*Hughes, 1989*), members of a given species of sea anemone can exhibit different life histories, as different as clonal versus solitary, in response to a combination of genetic and environmental variation (e.g., *Sebens, 1979*; *Sebens, 1980*; *Shick,*

*Hoffmann & Lamb, 1979*; *Bucklin, 1985*; *Lin, Chen & Chen, 1992*; *Tsuchida & Potts, 1994a*; *Tsuchida & Potts, 1994b*). In this study, we investigated a unique behavior of forced asexual reproduction in a sea anemone by its crab symbiont.

Our laboratory observations have shown that the *Lybia* larvae hatch from their egg without sea anemones, ruling out vertical transfer. It has been anecdotally reported (*Duerden, 1905*; *Karplus, Fiedler & Ramcharan, 1998*) that *Lybia edmondsoni* tear *T. producta* into two fragments, which later regenerate. *Karplus, Fiedler & Ramcharan (1998)* observed that if *Lybia* lose both sea anemones it may resort to intraspecific theft. Sea anemone theft has been documented both in intraspecific (*Giraud, 2011*) and interspecific (*Ross, 1979*) hermit crab confrontations. This behavior is very size dependent, whereby the larger of the two crabs will succeed in stealing a sea anemone (*Ross, 1979*; *Giraud, 2011*).

In the present study, we examined three hypotheses: (1) the pair of sea anemone held by a crab is an outcome of splitting a single sea anemone; (2) crabs deprived of sea anemones will steal a whole sea anemone, or fragment, from a conspecific organism; (3) these interactions affect the genotype structure of field populations of sea anemones.

To test these hypotheses, we conducted behavioral experiments intended on empirically testing the anecdotal reports of sea anemone ''splitting'' and intraspecific theft. In addition, we performed a genetic analysis using amplified fragment length polymorphism (AFLP; *Vos et al., 1995*) of sea anemone pairs held by *L. leptochelis* right after collection from the sea to assess the genetic relationship between each pair and to the population as a whole. AFLP is an efficient, fast and low cost DNA fingerprinting method (*Bensch & Åkesson, 2005*; *Meudt & Clarke, 2007*), particularly when studying organisms with limited prior knowledge of their genome (*Uthicke & Conand, 2005*). In addition, there is an increasing interest in the use of AFLP on marine invertebrates (*Uthicke & Conand, 2005*; *Peng et al., 2012*; *Goncalves et al., 2014*), particularly cnidarians (*Amar et al., 2008*; *Reitzel et al., 2008*; *Chomsky et al., 2009*; *Douek, Amar & Rinkevich, 2011*; *Brazeau, Lesser & Slattery, 2013*). If the crabs in nature behave like those observed in the laboratory, namely, frequent ''splitting'' and theft of sea anemones, we would expect to see high levels of genetic identity between each sea anemone pair. The ultimate aim of this study is to explore splitting and intraspecific theft, which forces asexual reproduction, consequently leading to reduced genetic variability in sea anemones held by boxer crabs.

## MATERIALS AND METHODS

### Collection of animals

Individuals of *Lybia leptochelis* and their symbiotic sea anemones *Alicia* sp. were collected from the shallow infra littoral zone at two separated beaches in Eilat, Israel during 2007–08 and again during 2013. The sites were approximately 3 km apart, Tur-Yam (29°31′49.69N; 34°55′36.39″E) and Red Rock Beach (29°31′01.40″N; 34° 55′13.34″E). Only intact crabs were collected. Oviparous females were not collected. Female crabs were observed carrying eggs from >4 mm carapace width (CW; Y Schnytzer, 2008, unpublished data), and therefore female crabs at least this size were defined as adults. The collected crabs had a CW between 5 to 11 mm. The sea anemones held by the crabs were ≤2.5 mm pedal disc diameter (PDD).

Using a small hand-held net, the crabs were collected and then individually placed in 0.5 L bottles filled with fresh sea water from the collection site, kept in a thermally insulated box and transported to Bar-Ilan University, Ramat Gan, Israel. The animals were collected and maintained within the guidelines of the Israel Nature and National Parks Authority (Permit no. 26103/2006/7/13).

## Sea anemone removal

For the splitting experiment, each of the crabs had one sea anemone removed. For the theft experiment both sea anemones from half of the crabs were removed. The removal process was based on the protocol presented by *Karplus, Fiedler & Ramcharan (1998)*. The crab was held in a glass Petri dish with enough sea water to cover it. The crab was then placed under a binocular microscope for constant monitoring. A solution of 7.5% $MgCl_2$ in distilled water was used to relax the sea anemones and prevent their contraction during removal. The solution was pipetted into the Petri dish in 500-µL increments every 2 min. Removal of the sea anemones took between 50 and 80 min. On some rare occasions, it was possible to remove the sea anemone from the crab's claws without $MgCl_2$ sedation. All the crabs, including those that did not have their sea anemones removed, were treated equally by the crab handler (i.e., sedation and contact with delicate forceps) to control for possible effects of crab "harassment."

## Animal measurement

The crabs with and without sea anemones and the lone sea anemones were photographed in small Petri dishes half filled with water placed on millimeter paper. The sea anemones were photographed after settling on the bottom of the dish. The CW of each crab was measured from the two furthest points on each side of the carapace (anterolateral lobes), and the PDD of each sea anemone was measured using Image J (NIH freeware) software.

## Experimental set up-general

All the crabs used for the behavioral experiments were individually maintained in the laboratory in small seven liter aquaria. Each aquarium was provided with a standard corner filter and a 5 cm long black PVC pipe lengthwise cut, which served as a shelter. The crabs and their sea anemones were fed every two days *ad libitum* with frozen adult *Artemia*. For further details of the general setup, day/night lighting regime, temperature and water quality in the aquaria see *Schnytzer et al. (2013)*.

## Crab sea anemone field data

Over the course of three years we documented the size of 54 *L. leptochelis*, 22 male and 32 females, which sea anemones they held and their size. We measured the crab and sea anemone sizes (as detailed above) right after collection from the sea.

## Sea anemone splitting experiment

To empirically test the hypothesis that when left with one sea anemone, *L. leptochelis* will split the other, we conducted the following experiment: twenty two *L. leptochelis* (14 males and eight females) had one sea anemone removed (as detailed above). We performed this for both left (10 trials) and right (12 trials) held sea anemones. Upon removal of one sea

anemone, the crab was placed in a small aquarium (18 × 10 × 10 cm) and monitored with a video camera (VHS HI8; Sony or Lumix TS2; Panasonic) for a period of 2–3 h. The trials were conducted in a closed room, behind a black curtain in order to minimize human interference. In the event that the crab split the sea anemone within this time frame, the trial was terminated and the crab was returned to its normal holding aquarium. In the event that the crab did nothing, the video recording was terminated after three hours and the crab was returned to its normal holding aquarium. However, the crabs that did not split the sea anemone in the initial monitoring period were examined twice a day for a period of two weeks. In any event of splitting, the crabs and their sea anemones were measured 10–14 days after the splitting and their morphology was assessed for regeneration (base, column, mouth and tentacles). See above section for measurement details.

## Sea anemone theft experiment

To assess the stealing behavior of *L. leptochelis*, 44 specimens of *L. leptochelis* were grouped into 22 pairs, comprising of crabs of similar size and same gender (14 male pairs and eight female pairs; new cohort, not crabs used in previous splitting experiments). The crabs ranged in size from 4–10 mm CW, with a maximal difference of 0.3 mm between each pair. Male–female pair trials were conducted during the preliminary stages of the study. Their behavior was identical to same sex pairs. However, following sea anemone theft/attempts, the fight was often followed by mating. Thus, to avoid confounding behavioral factors, only same sex trials were conducted. Each crab was only tested once. Each pair consisted of one crab holding both of its sea anemones, and the other had both removed. The crabs without sea anemones had them removed between two to five days prior to the contest. All the crabs were handled in the same manner, even if sea anemones were not removed, to control for the harassment effect. White Styrofoam boards placed between each crab aquarium prevented the crabs from coming into visual contact with their conspecifics. A black canvas sheet was hung over the experimental setup, minimizing the visual contact between the observer and the animals. The rest of the holding conditions were as mentioned above. The contests were conducted in part under daylight conditions (14 trials), and in part under night conditions (eight trials). The night trials were conducted under a dim red light, as it does not appear to have an effect on their behavior (*Schnytzer et al., 2013*). In general, *Lybia* crabs are more active at night (*Karplus, Fiedler & Ramcharan, 1998*; Y Schnytzer, 2008, unpublished data). However, during the preliminary stages of this study we observed that the crabs were equally active when placed into the same small aquarium, so the trials were conducted under both light regimens. In the trials conducted under daylight conditions, identification of the individual crabs was conducted based on observable differences in their coloration. For the night trials, the crabs were marked with a small piece of plastic affixed to the dorsal surface of their carapace with a cyano-acrylate ester based adhesive (Super Glue).

In each trial, two crabs were introduced into an aquarium (23 × 23 × 20 cm), each inside a separate transparent glass cylinder on opposing sides of the aquarium. After 10 min of acclimation, the cylinders were slowly and simultaneously removed. In the event that no contact was made between the crabs after a period of 45 min the trial was terminated. The
behavioral interactions between the crabs were recorded with a digital video camera (VHS HI8; Sony or Lumix TS2; Panasonic). Typically, during the preliminary trials, we observed that after coming into contact, whether theft occurred or not, each crab would retreat into a corner of the aquarium and no longer approached the other, thus the trials were terminated at this stage. At the end of each trial, the crabs were returned to their original aquaria for a period of two weeks. During this period, daily observations were made for the monitoring of sea anemone splitting activity.

## DNA extraction

For AFLP analysis, DNA was extracted from fresh material. Genomic DNA was extracted using a High Pure PCR Template Preparation Kit (Roche, Mannheim, Germany) according to the manufacturer's protocol. Due to their small size, DNA was extracted from the entire sea anemone. DNA concentration was determined by a NanoDrop ND1000 (Thermo Fisher Scientific Inc., Waltham, MA, USA) at 260 nm.

## Amplification and "fingerprinting"

Eight pairs of sea anemones removed from *L. leptochelis* from the two above mentioned Eilat beaches were analyzed (specimens 1–5 from Tur-Yam; 6–8 from Red-Rock Beach). We employed AFLP genotyping (*Vos et al., 1995*) with modifications according to *Huys & Swings (1999)* and *Amar et al. (2008)*, in which radioactive labeling was replaced with fluorescent dyes. Restriction enzyme digests were performed on 250 ng of genomic DNA for 3 h at 37 °C using two restriction enzymes (MseI and EcoRI), followed by the ligation of respective double strand adapters (EcoRI adaptor E1-CTCGTAGACTGCGTACC and E2-AATTGGTACGCAGTCTAC, and MseI adaptors M1-GACGATGAGTCCTGAG and M2-TACTCAGGACTCAT). The E1 and M1 oligonucleotides were used as primers for pre-selective PCR amplification using 1 μl of ligation products for the second selective amplification. The PCR product was diluted 1:50, and 5 μl was used for the second amplification. We used three pairs of fluorescent labeled primers (VIC, FAM, and NED; Applied Biosystems, Foster City, CA, USA) as follows: (E=GACTGCGTACCAATTC+XXX and M=GATGAGTCCTGAGTAA+XXX): VIC—E+ACC: 5′ 3′with M+CTC: 5′ 3′; NED—E+ACA: 5′ 3′with M+CTC: 5′ 3′; and FAM—E+AGC: 5′ 3′with M+CTT: 5′ 3′. The process was repeated twice (duplicates) for each sample to attain maximum accuracy.

## AFLP analysis

DNA sequencing was performed at the Instrumentation and Service Center of the George S. Wise Faculty of Life Sciences, Tel-Aviv University. The samples were analyzed using a Genetic Analyzer 3100 (ABI PRISMA; Applied Biosystems). The samples were diluted and 0.3–0.5 μl of size standard Lis 600 was added to the PCR product in the presence of formamide. Fluorescent-labeled PCR products appear as peaks and were first analyzed using GeneScan ABI PRISM 3.7 software (PE Biosystems; *Oda et al., 1997*) to determine peak sizes in base pairs, according to the size marker. Each PCR peak obtained from the samples was then aligned and converted into a binary system. The results were transferred to binary scores (0, 1) using AFLP Macro2 software. Nei's genetic distance (*Nei, 1978*) was calculated using POPGENE version 1.31 (http://www.ualberta.ca/~fyeh). The binary

results were then converted to NEXUS format and the maximum parsimony option of PAUP was used to build a dendrogram of the sea anemone population.

## Statistical analysis

The sea anemone asymmetry index represents the relative difference in the pedal disc diameter of the two sea anemones held by a crab, either directly from the sea or those split in the lab. The sea anemone asymmetry index (*Ianem*) was calculated by subtracting the pedal disc diameter of the smaller sea anemone (PDDs) from the larger one (PDDb) and dividing the difference by the larger sea anemone pedal disc diameter.

$$Ianem = (PDDb - PDDs)/PDDb$$

Correlation analyses between field collected crabs and sea anemone sizes (CW/mm for crabs and PDD/mm for sea anemones) was conducted by using a Pearson's product moment correlation test. A Welch two sample $t$-test was used to test for differences between the size of male and female held sea anemones, to test whether or not gender has an effect on the asymmetry index. Binomial probability tests were carried out on the splitting and theft scores to determine whether the proportion of outcomes differed significantly from the expected 50% chance level. In the splitting experiment, multiple linear regression analyses were performed to assess the effect of crab gender, sea anemone size, and handedness on the time duration from the moment a sea anemone was removed until the remaining one was split. Further multiple regressions were done to test whether the asymmetry index was predicted by crab gender, time to split, sea anemones size and handedness. In the theft experiment, a multiple regression was performed to test if crab gender and fight outcome had an effect on fight duration. A two-way ANOVA was performed to test whether crab gender, initiator of fight (with or without sea anemones), or the interaction between them, had an effect on lag to start of fight. In case of non-normal distribution, data were log transformed. Data were checked for normality using a Kolmogorov–Smirnov test. All statistical tests used in this study employed a significance level of $\alpha = 0.05$. The analyses were conducted using SPSS 15.0 or R (https://www.r-project.org/).

# RESULTS

## Crab-sea anemone field measurements

During the course of this study, all *L. leptochelis* collected or observed in nature, well over one hundred specimens, were found holding a pair of *Alicia* sp. (Fig. 1). The sea anemones held in the left and right claws are significantly correlated in size (Pearson's product-moment correlation, $r = 0.90$, $t_{52} = 14.883$, $P < 0.0001$; Fig. 2). In addition, the sea anemones significantly correlate to the size of the crab holding them (Pearson's product-moment correlation, $r = 0.72$, $t_{52} = 7.4546$; $P < 0.0001$; Fig. 3). There is a highly significant difference between the size of sea anemones held by males (X $\pm$ SD = 1.37 $\pm$ 0.51 PDD/mm) vs. females (X $\pm$ SD = 1.92 $\pm$ 0.57 PDD/mm; Welch two sample $t$ test; $t_{42.156} = 3.6513$, $P < 0.001$). In contrast, gender had no effect on the asymmetry index (male: X $\pm$ SD = 9.69 $\pm$ 9.39%, female: X $\pm$ SD = 12.85 $\pm$ 10.22%; Welch two sample $t$ test, $t_{42.73} = 1.1513$, $P = 0.256$).

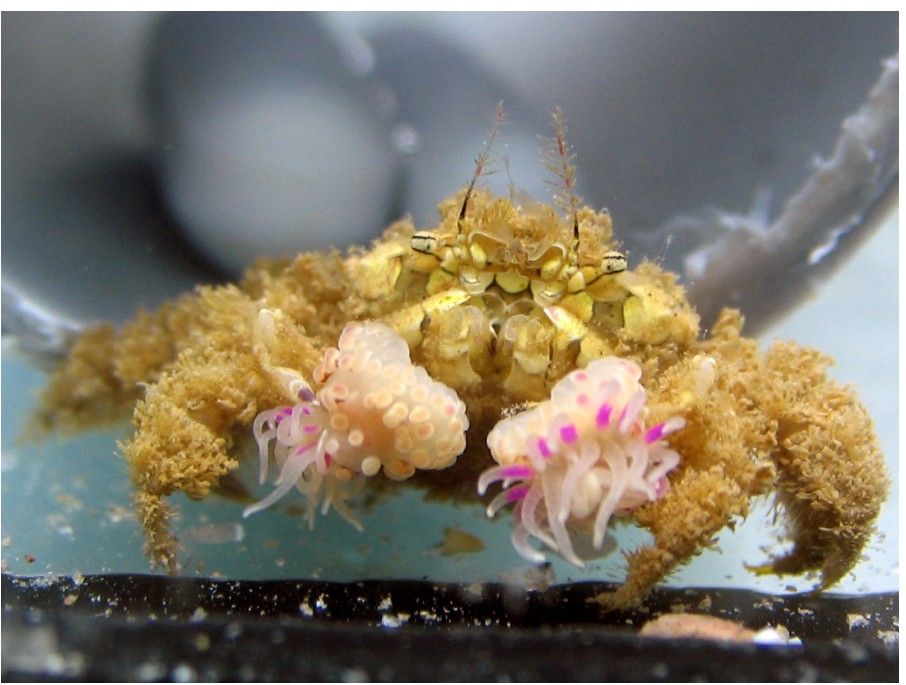

**Figure 1** *Lybia leptochelis* collected directly from the sea holding typically similar sized *Alicia* sp. anemones.

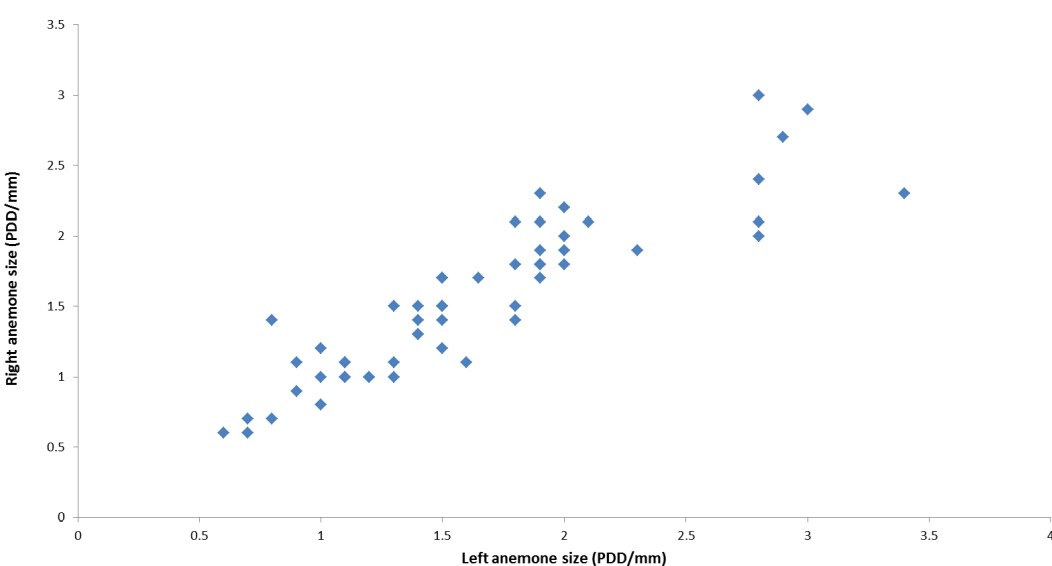

**Figure 2** **Correlation between left and right held anemones as observed in nature.** *Lybia leptochelis* hold significantly similar sized *Alicia sp.* anemones in each claw. ($r = 0.90$, $t_{52} = 14.883$, $P > 0.0001$). PDD, pedal disc diameter measurements are in mm.

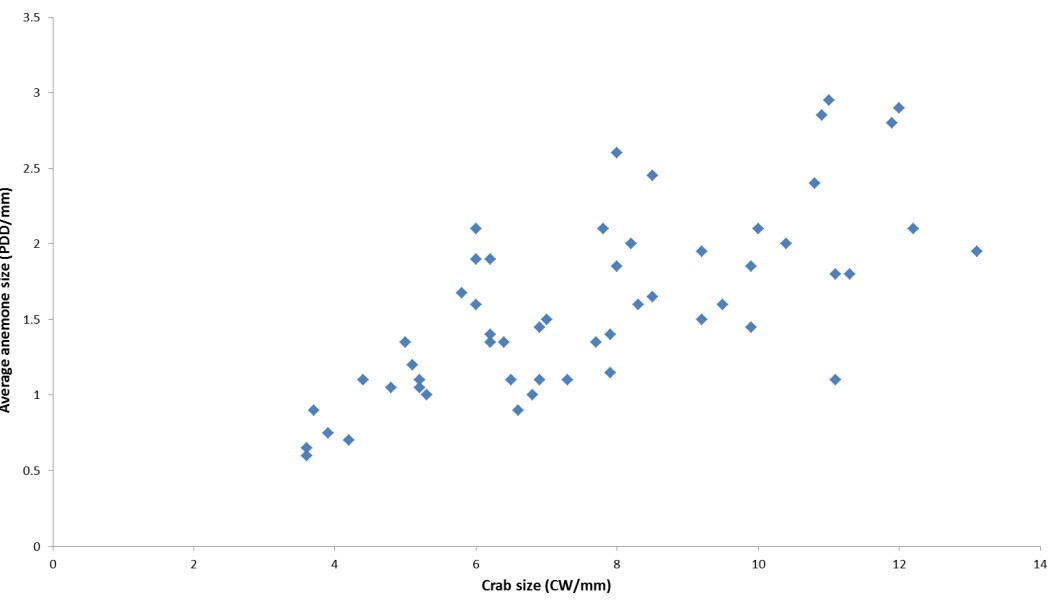

**Figure 3** **Correlation between held anemones (average of left and right anemones) and crab size as observed in nature.** ($r = 0.72$, $t_{52} = 7.4546$, $P < 0.0001$). PDD, pedal disc diameter; CW, Carapace width measurements are in mm.

## Sea anemone splitting experiment

Seventeen out of the twenty two crabs holding a single sea anemone split it within six days after the removal of one of their two sea anemones. The sea anemones were split into two clones which subsequently regenerated into two intact sea anemones (Table 1). The splitting behavior was a highly significant response to sea anemone removal, performed in 77% of the trials (binomial test, $P = 0.02$, $N = 22$). The five crabs that did not split their sea anemones within the two week duration of the experiment were composed of both large and small individuals of both genders and their single sea anemone pedal disc diameter overlapped with that of sea anemones which were split by the crabs (Table 1).

Time from the removal of one sea anemone until the splitting of the remaining one was highly variable, ranging from one hour to six days with a mean ($\pm$SD) of $29.2 \pm 35.2$ h until sea anemone splitting. Time to split was not well predicted by crab gender (males: $X \pm SD = 25.5 \pm 41.32$ min; females: $X \pm SD = 36.0 \pm 21.5$ min), sea anemone size or handedness (multiple linear regression; Table 2A).

The actual splitting process was observed several times in its entirety and lasted between 1 min and over 2 h. Typically, the actual process of splitting lasted approximately 20 min, taking the following course: the crab held the sea anemone across the column, with the pedal disc facing upward and the oral disc and tentacles facing downward. The crab then took hold of the sea anemone with its free claw, thus holding the sea anemone in the aforementioned downward conformation between both claws (Figs. 4A and 4B; Video S1). Next, the crab slowly began stretching the sea anemone between both claws in an outward motion, utilizing its front walking legs in order to surgically tear the sea anemone in half (Fig. 4C). Occasionally, the crab momentarily ceased the stretching to re-grasp the sea

**Table 1  Anemone splitting by *Lybia leptochelis* following removal of one of its anemones.**

| Crab number | Crab gender | Crab carapace width (mm) | Remaining anemone size; held by left (L) or right (R) claw | Anemone splitting | Time to split (h) | Size of Anemone held in right claw 10–14 days following splitting | Size of Anemone held in left claw 10–14 days following splitting | Asymmetry index |
|---|---|---|---|---|---|---|---|---|
| 1 | F | – | 0.9 (R) | – | – | 1.1 | – | – |
| 2 | M | 4.2 | 1.0 (L) | + | 4 | 0.8 | 0.8 | 0 |
| 3 | M | 4.1 | 1.1 (L) | + | 4 | 0.8 | 0.8 | 0 |
| 4 | F | 4.7 | 1.1 (R) | + | 48 | 0.8 | 0.8 | 0 |
| 5 | M | 4.5 | 1.1 (L) | + | 36 | 1.0 | 0.9 | 10% |
| 6 | M | 4.4 | 1.1 (L) | – | – | – | 1.1 | – |
| 7 | M | 4.7 | 1.2 (R) | + | 4 | 0.8 | 0.8 | 0 |
| 8 | M | 4.7 | 1.2 (R) | + | 4 | 0.8 | 0.8 | 0 |
| 9 | M | 4.1 | 1.3 (R) | + | 36 | 1.1 | 0.9 | 18% |
| 10 | F | 6.3 | 1.3 (L) | + | 36 | – | – | – |
| 11 | F | – | 1.4 (L) | – | – | – | 0.9 | – |
| 12 | M | – | 1.4 (L) | + | 12 | – | – | – |
| 13 | M | 8.0 | 1.5 (L) | + | 12 | 1.3 | 1.2 | 7.7% |
| 14 | M | – | 1.6 (R) | – | – | 1.8 | – | – |
| 15 | F | – | 1.6 (R) | + | 24 | 1.4 | 1.2 | 14.3% |
| 16 | M | – | 1.6 (R) | + | 144 | 1.1 | 0.9 | 18% |
| 17 | F | – | 1.7 (R) | + | 24 | 1.5 | 1.1 | 26.6% |
| 18 | F | 10.1 | 1.8 (R) | + | 72 | 1.1 | 1.2 | 8.3% |
| 19 | M | 8.6 | 1.8 (R) | – | – | 2.0 | – | – |
| 20 | M | 8.0 | 2.0 (L) | + | 1 | 1.4 | 1.3 | 7.1% |
| 21 | M | – | 2.1 (R) | + | 24 | – | – | – |
| 22 | F | – | 2.5 (R) | + | 12 | – | – | – |

Notes.
+, Crab split anemone.

anemone in what appears to be the most centered conformation possible, so that the final splitting will produce two equal parts. Once the sea anemone has been re-grasped, the crab initiated the stretching once again, slowly pulling the sea anemone from the center outwards. Once the majority of the sea anemone was split into two, there were often final strands of sea anemone tissue connecting each newly split sea anemone, which were torn by the front walking legs (Figs. 4D and 4E). Once the splitting process was complete the crab had two identical clones held in each claw (Fig. 4F).

Overall, following splitting and sea anemone regeneration, the pedal disc surface area of the two new sea anemones increased substantially (X ± SD = 10.0 ± 23.2%) in comparison to the single sea anemone prior to splitting. However, in some cases, the combined pedal disc surface area of the two new sea anemones was similar or even smaller than that of the original sea anemone. This phenomenon is reflected in the large standard deviation of the increase in pedal disc surface area following splitting.

**Table 2** Multiple linear regression model of (A) Time to split and (B) Asymmetry index.

| Factor | Coefficient | SE | $t$ |
|---|---|---|---|
| **(A) Time to split** | | | |
| Constant | 5.3369 | 2.3896 | 2.233 |
| Crab gender | −2.1798 | 1.1833 | −1.842 |
| Anemone size | −1.0365 | 1.2963 | −0.800 |
| Handedness | 0.2399 | 0.9300 | 0.258 |
| $F_{3,6} = 1.852, R^2_{Adj} = 0.2211, P = 0.2385$ | | | |
| **(B) Asymmetry index** | | | |
| Constant | −0.285792 | 0.067879 | −4.210[**] |
| Crab gender | 0.197162 | 0.042242 | 4.667[**] |
| Time to split | 0.003594 | 0.000705 | 5.098[**] |
| Anemone size | 0.072579 | 0.030737 | 2.361[^] |
| Handedness | 0.006401 | 0.022159 | 0.289 |
| $F_{4,5} = 8.351, R^2_{Adj} = 0.7657, P = 0.01941$ | | | |

**Notes.**
[**]$P < 0.001$.
[^]$P = 0.065$.

The sea anemone asymmetry index calculated for the two sea anemones resulting from the splitting process was overall small (X ± SD = 8.5 ± 8.7%). The asymmetry index ranged however from 0 to 26% reflecting the high value of the standard deviation of the calculated index. The multiple linear regression model (Table 2B) shows that both crab gender and time to split from sea anemone removal were significantly related to the asymmetry index, sea anemone size was weakly related, and handedness was unrelated.

## Sea anemone theft experiment

In 73% of the staged encounters between crabs with and without sea anemones, intense fighting took place, culminating in sea anemone theft (binomial test, $P = 0.05$, $N = 22$; Table 3). In 44% of the contests, an entire sea anemone was stolen, in 37% a sea anemone fragment was taken, and in the remaining 18% the crab without sea anemones came away with two sea anemone fragments (Table 3). Out of the six remaining trials which did not end in sea anemone theft, in five cases the crabs refrained from fighting, in two of them they did not move over a period of 45 min (no contact) and in the three remaining cases the crabs only made gentle leg contact before separating. The only trial in which there was aggressive contact, but no theft occurred was the shortest recorded contest (1:23 min) where the crabs mainly collided into each other but lacked the typical contest structure described below.

Crabs of both genders, with or without sea anemones, were equally likely to initiate a fight (binomial test, $P = 0.6291$, $N = 17$). Contests between crabs started on average 15.5 ± 6.5 min after the acclimation period. A two-way ANOVA revealed that neither crab gender (males: X ± SD = 16.2 ± 7.5; females: X ± SD = 14.3 ± 4.3), nor whether the initiator was deprived or in possession of sea anemones had a significant effect on time until the start of fighting (Table 4).

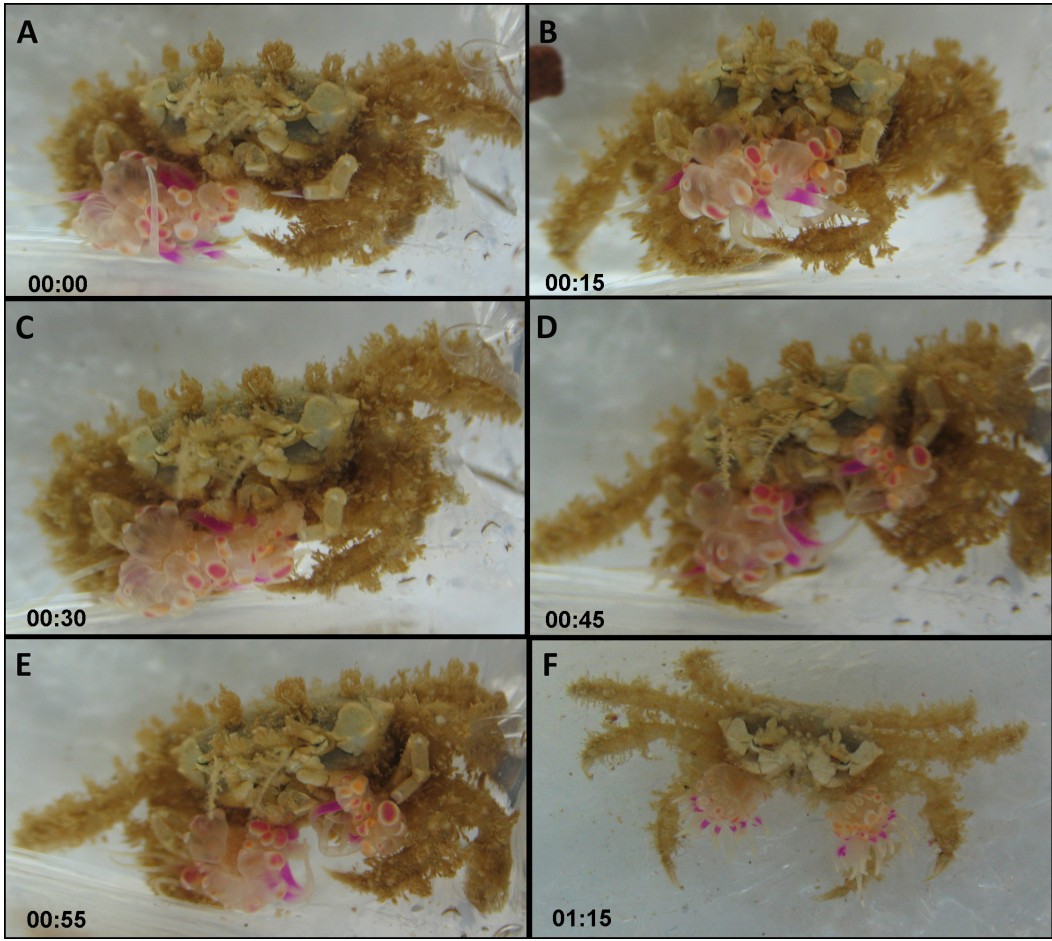

**Figure 4  Sequence of anemone splitting behavior.** This particular trial took approximately 1.2 h until splitting was completed. Time presented in hh:mm format. (A) *Lybia leptochelis* holding an *Alicia* sp. in one claw the second is vacant. (B) Typical anemone splitting conformation with pedal disc up and oral disc/tentacles down. (C) Stretching of the anemone between both claws and use of front walking legs to tear it down the middle. (D) Tearing of anemone into two. (E) Final strands of anemone tissue are cut with front walking legs. (F) *L. leptochelis* holding two identical clones of the original *Alicia* sp. anemone.

Typically, after being placed together and the cylinders removed, one contestant would approach the other. For the sake of illustration, we will describe a crab with sea anemones approaching one without them. As the crab with sea anemones came within a close proximity of the crab without sea anemone, the crab with sea anemones held its sea anemones at a distance away from the other crab (Fig. 5A). Next, the initiator gently touched the other crab with the tip of its first walking leg for about a minute (Fig. 5B). Following this gentle leg contact, the two crabs typically proceeded to move into a back to back configuration (Fig. 5C). Following this, the crabs rapidly locked their walking legs and commenced a close physical struggle grasping one another with their legs forming a tight ball. It is important to note that during these phases both crabs distanced their claws (holding sea anemones or vacant) as far as possible from the other (Fig. 5D). Next, the crab without sea anemones strived to move into a dominant position, typically on top of

**Table 3  Theft of anemones during encounters between *Lybia leptochelis* with and without sea anemones.**

| Crab pair number and gender | Fight initiator | Minutes till beginning of fight | Fight duration (min) | Fight outcome | Splitting |
|---|---|---|---|---|---|
| 1 F | +A | 17 | 40 | Theft of an anemone fragment. | + |
| 2 F | +A | 11 | 31 | Theft of a complete anemone. | + |
| 3 F | – | – | – | No theft. | – |
| 4 M | +A | 21 | 32 | Theft of a complete anemone. | + |
| 5 M | – | – | – | No theft. | – |
| 6 M | – | – | – | No theft. | – |
| 7 M | – | – | – | No theft. | – |
| 8 F | – | – | – | No theft. | – |
| 9 F | −A | 12 | 14 | Theft of an anemone fragment. | + |
| 10 M | −A | 15 | 32 | Theft of two anemone fragments. | – |
| 11 M | −A | 19 | 32 | Theft of a complete anemone. | + |
| 12 M | +A | 12 | 25 | Theft of an anemone fragment. | + |
| 13 F | −A | 12 | 6.5 | Theft of a complete anemone. | + |
| 14 M | −A | 1.66 | 3.66 | Theft of a complete anemone. | + |
| 15 M | −A | 19 | 1.23 | No theft. | – |
| 16 F | +A | 12 | 1.66 | Theft of an anemone fragment. | + |
| 17 M | +A | 12 | 17 | Theft of two anemone fragments. | – |
| 18 M | +A | 20 | 12 | Theft of an anemone fragment. | + |
| 19 M | −A | 32 | 7.5 | Theft of an anemone fragment. | + |
| 20 M | +A | 12 | 13 | Theft of a complete anemone. | + |
| 21 F | +A | 22 | 10 | Theft of a complete anemone. | + |
| 22 M | +A | 15 | 20 | Theft of two anemone fragments. | – |

**Notes.**
+A, Crab holding anemones; −A, Crab without anemones.

**Table 4  Two-way ANOVA investigating the effect of crab gender and fight initiator on time until start of fight.**

| Source of variation | df | Sum | F | P |
|---|---|---|---|---|
| Constant | 1 | 288.0 | 5.89 | 0.031 |
| Crab gender | 1 | 40.61 | 0.83 | 0.379 |
| Initiator | 1 | 16.33 | 0.33 | 0.573 |
| Crab gender × initiator | 1 | 27.08 | 0.55 | 0.470 |
| Error | 13 | 636.1 | | |

**Notes.**
Initiator, Crab with or without anemones.

the crab holding sea anemones (Fig. 5E). The crab without sea anemones then proceeded to try and hold one of the opposing crab's claws and lock it with the aid of its walking legs. No use was made of its unoccupied delicate claws (Fig. 5F). Upon achieving a claw lock of the opposing crab (Fig. 5G), the crab without sea anemones proceeded to remove the sea anemone held by the other crab. At first, it made use of its first walking leg to pry at the claw holding the sea anemone. After it has been pried open sufficiently, the attacking crab for the first time used its vacant claw to take hold of the sea anemone (Fig. 5H). Sometimes,

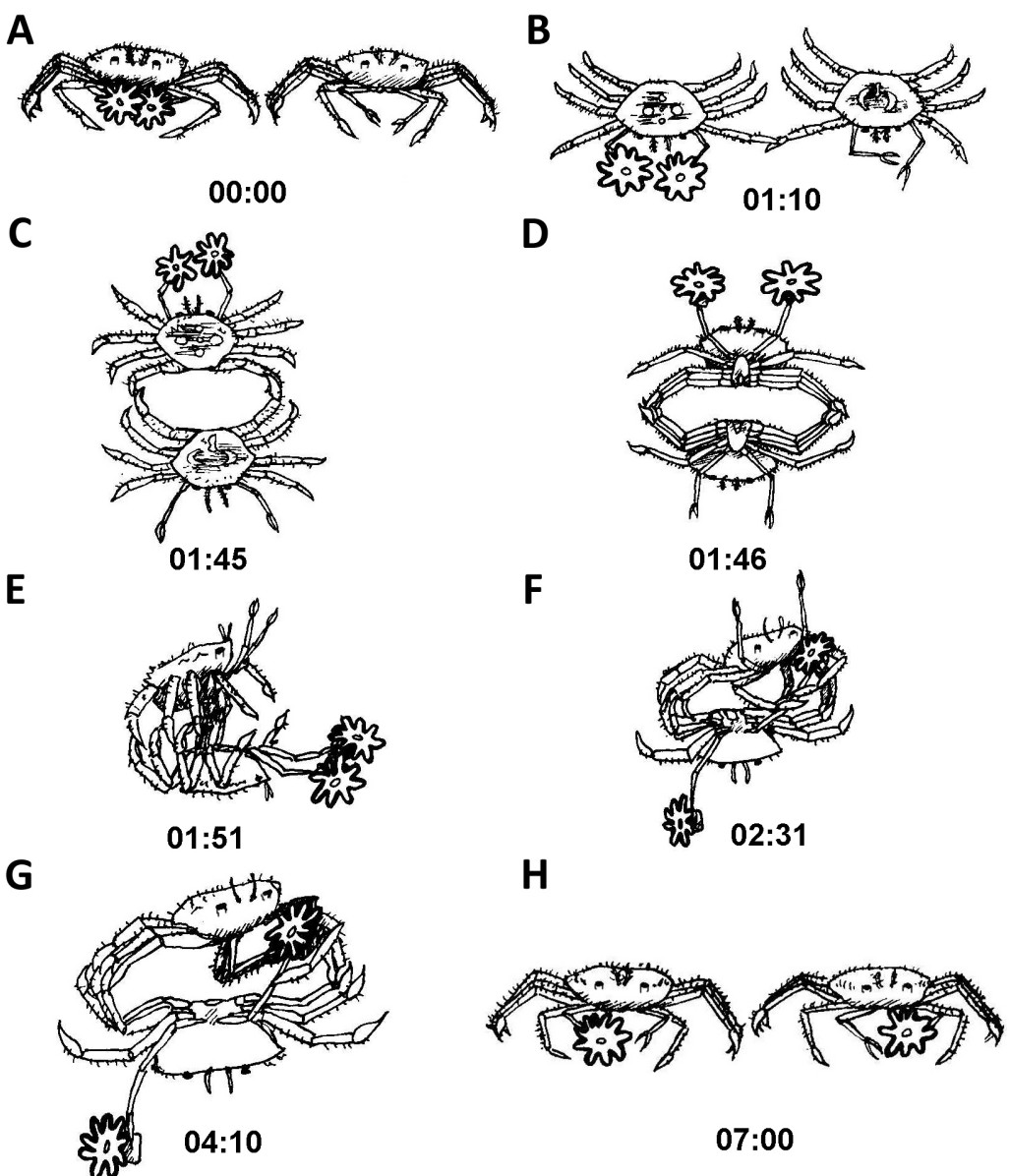

**Figure 5 Sequence of anemone theft behaviour line drawing from video.** Time presented in mm:ss. Please refer to 'Results' for elaboration on theft sequence.

an entire sea anemone was taken and sometimes only a fragment was torn off. We never witnessed a contest where two whole sea anemones were stolen. Typically, after a whole or partial sea anemone has been taken the contestants broke off and "returned to their corners" (Video S2).

The fight duration was extremely variable, ranging from between less than a minute to 40 min with average ($\pm$SD) durations of $17.5 \pm 12.4$ min per fight. A multiple linear regression model failed to show a connection between fight duration and crab gender (males: X $\pm$ SD = $17.7 \pm 11.5$; females: X $\pm$ SD = $17.2 \pm 15.0$) or contest outcome (i.e., removal of a complete sea anemone or a sea anemone fragment) (Table 5). Sea

**Table 5  Multiple linear regression model of fight duration.**

| Factor | Coefficient | SE | t |
|---|---|---|---|
| Constant | 16.4062 | 6.4844 | 0.0264 |
| Crab gender | 0.5742 | 7.3154 | 0.9387 |
| Complete anemone | 1.5742 | 7.3154 | 0.8332 |
| Two fragments | 6.0196 | 9.9692 | 0.5572 |

$F_{3,12} = 0.158$, $R^2_{Adj} = -0.2025$, $P = 0.923$

**Table 6  Pairwise unbiased (_Nei, 1978_) genetic identities (above diagonal) and genetic distances (below diagonal) between _Alicia sp._ anemone removed from single crab.** Collection location and pair of crab are indicated: Ty, Tur Yam; Rr, Red rock; each capit.

| Pop ID | RrH | TyB | TyC | TyD | TyE | RrF | RrG | TyA |
|---|---|---|---|---|---|---|---|---|
| RrH | | 0.7931 | 0.7931 | 0.7931 | 0.7931 | 0.7931 | 0.7931 | 0.8759 |
| TyB | 0.2318 | | 1.0000 | 1.0000 | 1.0000 | 1.0000 | 1.0000 | 0.8759 |
| TyC | 0.2318 | 0.0000 | | 1.0000 | 1.0000 | 1.0000 | 1.0000 | 0.8759 |
| TyD | 0.2318 | 0.0000 | 0.0000 | | 1.0000 | 1.0000 | 1.0000 | 0.8759 |
| TyE | 0.2318 | 0.0000 | 0.0000 | 0.0000 | | 1.0000 | 1.0000 | 0.8759 |
| RrF | 0.2318 | 0.0000 | 0.0000 | 0.0000 | 0.0000 | | 1.0000 | 0.8759 |
| RrG | 0.2318 | 0.0000 | 0.0000 | 0.0000 | 0.0000 | 0.0000 | | 0.8759 |
| TyA | 0.1325 | 0.1325 | 0.1325 | 0.1325 | 0.1325 | 0.1325 | 0.1325 | |

**Notes.**

Ty, Tur Yam; Rr, Red rock.

anemone splitting was observed in all instances where a complete or a fragment of the sea anemone was stolen (Table 3). In the event that two fragments were stolen, splitting was not observed.

## AFLP

Sixteen sea anemones from eight crabs were analyzed. The three sets of fluorescent labeled primers (VIC, NED, and FAM) revealed 43, 30 and 71 bands, respectively (total = 144 bands). The sizes of the amplified fragments ranged between 60 to 430 bps. The majority of the bands were monomorphic, and only 24.9% (FAM = 15.5%; NED = 26.6%; VIC = 32.5%) were polymorphic. The fingerprint profiles of all sea anemone pairs taken from a single crab were identical. Between the pairs, six out of the eight sea anemones pairs (four from Tur-Yam and two from Red-Rock) were identical, and the two other pairs (one from Tur-Yam and the other from Red-Rock) exhibited independent banding patterns from the other six. A maximum parsimony dendogram (Fig. 6), as well as _Nei_'s (_1978_) mean genetic distance analyses (Table 6), revealed the presence of three genotypes.

## DISCUSSION

### Symbiotic sea anemones

_L. leptochelis_ from the Gulf of Eilat represent a unique case among _Lybia_ crabs with regard to their symbiotic sea anemones. Most _Lybia_ crabs are found holding the sea anemone _Triactis producta_, while _L. leptochelis_ from the Gulf of Eilat hold a pair of _Alicia sp._

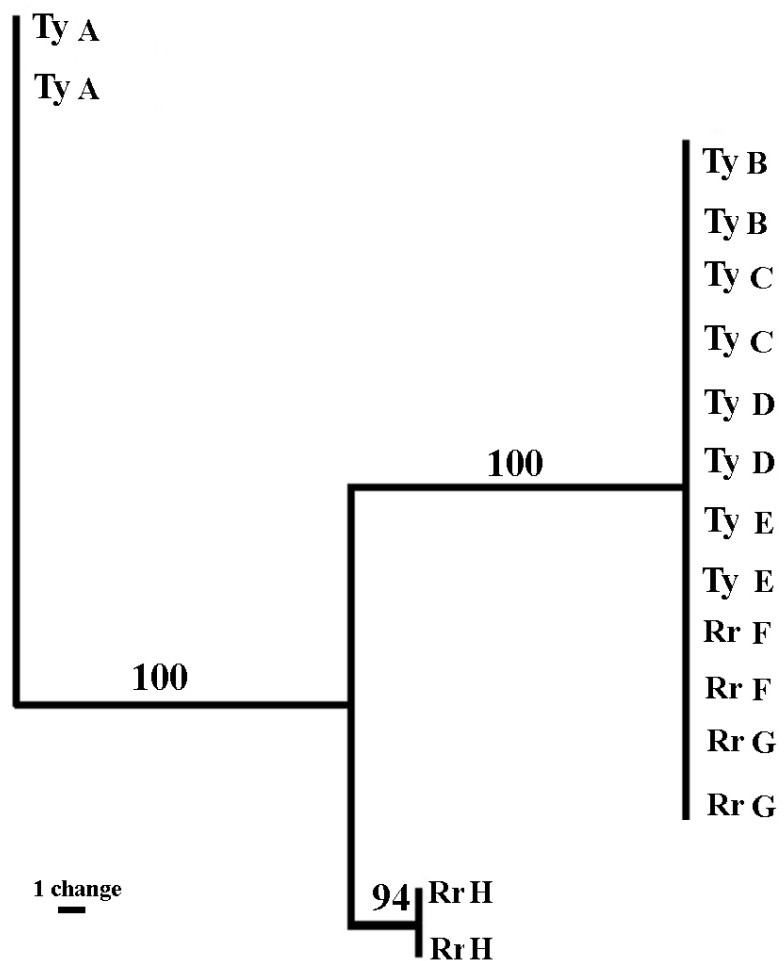

**Figure 6 Genetic relatedness of *Alicia* sp. anemone pairs taken from *Lybia leptochelis*.** The Maximum Parsimony dendogram is the combined results of 3 different primer combinations. Collection location: Ty, Tur Yam; Rr, Red rock. Each pair of letters represents a pair of anemones originating from a single *L. leptochcelis*.

(*Schnytzer et al., 2013*). Free living specimens of this unidentified species of *Alicia* were not found in or around the crab's habitat over the course of some four years of research in the area, and there is no previous description of them in the literature.

All crabs found in nature during this study were holding a pair of *Alicia* sp. Even the smallest crabs found (2 mm CW), probably not long after the megalopa settled, already possessed a pair of minute sea anemones. In a previous study we showed that *L. leptochelis* steal food from their held sea anemones, thus regulating their growth and subsequent size (*Schnytzer et al., 2013*). Indeed, there is a significant correlation between crab and sea anemone sizes (Fig. 3), suggesting an optimal carrying size. Females hold significantly larger sea anemones than males of similar size. Currently, we can only speculate on the nature of this 'sexual dimorphism.' Perhaps it aids in heightened protection provided by larger sea anemones, an evident advantage to egg carrying females. *Lybia* crabs are presumably obligatory symbionts of their held sea anemones as most previous studies also

report that all wild caught crabs were found holding a pair of sea anemones (*Duerden, 1905*; *Karplus, Fiedler & Ramcharan, 1998*; *Yanagi & Iwao, 2012*; *Schnytzer et al., 2013*). Only one study reported collecting several *L. tessellata* without sea anemones (*Borradaile, 1902*). Interestingly, the sea anemone most commonly associated with *Lybia* crabs (*Karplus, Fiedler & Ramcharan, 1998*; *Yanagi & Iwao, 2012*), *T. producta*, is found freely living in Eilat (*Fishelson, 1970*; Y Schnytzer, pers. obs., 2010), yet they have never been observed in association with *L. leptochelis*. Similar to our case, *Duerden (1905)* reported that the sea anemones, *Sagartia* and *Bunodeopsis*, held by *Melia tessellate* (=*Lybia edmondsoni*; see *Ross, 1974*) were not found freely living around the crabs habitat, during a careful search made over the course of three months. The apparent "rarity" of *Alicia sp.* in conjuncture with all the observed crabs holding sea anemones gives rise to the question of how they obtain them. Of course we cannot rule out the possibility that the sea anemones do occur in or around the crab's habitat and have yet to be found.

## Sea anemone splitting

*Duerden (1905)* and later *Karplus, Fiedler & Ramcharan (1998)* provided anecdotal evidence of *Lybia* crabs splitting sea anemones. We have empirically shown for the first time that in the vast majority of cases, a crab which has one sea anemone removed will split the other into two new ones. As our data show, this behavior appears to be independent of any crab or sea anemone physical characteristics, suggesting this is a dominant and widespread behavior. By splitting a sea anemone, the crab effectively induces asexual reproduction of the sea anemone. Indeed, all split sea anemones were observed fully regenerated within a matter of days. Consequently, sea anemone splitting appears to be a well-orchestrated behavior, conducted with apparent care for the final outcome, i.e., two new viable sea anemones (Fig. 4 and Video S1). Fission, i.e., programmed physical separation, is a well-known form of sea anemone asexual reproduction (*Geller, Fitzgerald & King, 2005*; *Sherman & Ayre, 2008*). However, as the classic definition implies, this is usually a self-regulated form of asexual reproduction. To our knowledge, there are no other known examples of other marine organism which physically induce this behavior in sea anemones. Commonly, animals associated with sea anemones will either reside around or within them, such as clownfish (*Karplus, 2014*) or a wide range of crustaceans (*Jonsson, Lundalv & Johannesson, 2001*; *Duris et al., 2013*; *Fernandez-Leborans, 2013*). Alternatively, animals which carry sea anemones on them, such as hermit crabs (*Williams & McDermott, 2004*) will either locate them freely living or engage in interpecific and intraspecific theft (*Ross, 1979*; *Giraud, 2011*; see below for further details). Crustaceans commonly place their associated sea anemone on their shell, carapace or walking legs (*Guinot, Doumenc & Chintiroglou, 1995*). The habit of physically holding sea anemones in their claws appears to be unique to the order Polydecdinae (*Duerden, 1905*; *Guinot, 1976*), a factor which may explain why this splitting behavior arose and is unknown among other crabs. Interestingly, our analysis revealed that time to split had a significant, albeit small, positive effect on the asymmetry index, indicating that a shorter time to split results in more equally sized sea anemones. Although this experiment was confined to laboratory conditions, we may cautiously assume that splitting sea anemones is presumably part of the crab-sea anemone

acquisition mechanism in nature (see AFLP results below). As mentioned above, our field data also supports this claim in that there is a highly significant correlation between sea anemone pair size held by crabs caught in the wild (Fig. 2).

## Sea anemone theft

There is only one report in the literature of sea anemone theft in *Lybia* crabs. *Karplus, Fiedler & Ramcharan (1998)* reported on one observation of sea anemone theft. This isolated case was observed when two small *L. edmondsoni* with sea anemones were introduced into an aquarium with a large conspecific deprived of sea anemones. Our experiment shows for the first time that this is a highly common behavior, occurring in the vast majority of instances, irrespective of sex, in which two crabs are placed together, one holding sea anemones and the other without. Interestingly, the initiation of contact was irrespective of sea anemone possession. One might have thought that this would be less likely due to the apparent "high value" of their sea anemones. These encounters exhibited a similar sequence of behaviors in all the trials we conducted. Upon initial contact, the initiator always "feels" the opponent's leg. In the three trials where contact was made but no fight was initiated, the crabs separated after this leg contact phase. Pre-fight assessment is a well-known behavior, often dictating whether or not animals will commence fighting (*Arnott & Elwood, 2009*). As can be seen quite clearly in the example video (Video S2), these battles are at times quite violent in their appearance. However, in no instances did we observe a crab being injured or killed. Unlike splitting, intraspecific sea anemone theft has been observed in hermit crabs, yet intraspecific theft is far less common. In hermit crabs, it is always the hermit crab lacking sea anemones which initiates the fight, and the larger of the two who prevails (*Ross, 1979*; *Ross, 1983*; *Giraud, 2011*). In many cases amongst crustaceans, there is a clear size advantage regarding resource acquisition (*Jaroensutasinee & Tantichodok, 2002*; *Pratt, McLain & Lathrop, 2003*; *Arnott & Elwood, 2009*). In contrast, this appears not to be the case with boxer crabs. *Lybia* crabs presumably acquire their sea anemones sometime after settling from the larval stage. Although quite difficult to find, we did manage to collect three tiny specimens (2–3 mm CW), and after removing their sea anemones we conducted three preliminary contests between them and fully grown crabs (8–10 mm CW). In all cases it was the small crab which initiated the fight, and in all instances it managed to come away with a sea anemone fragment or a full sea anemone (Video S3). As is evident in the video the small crab is quite determined to get a sea anemone, and despite the great difference in size it manages in much the same way as larger crabs to succeed. Although these are preliminary observations and under laboratory conditions, they are insightful into the possible mechanism of sea anemone acquisition in nature by small individuals. Following our observations in the splitting experiment, the crabs that stole a complete sea anemone or a fragment proceeded to split it up to two weeks after the contest. Interestingly, the crabs that stole two fragments, holding one in each claw, would not split. Presumably the instinct to split is not induced when both claws are occupied.

## AFLP

Genetic markers have been successfully used to determine the asexual origin of broods of sea anemone (*Schaefer, 1981*; *Carter & Thorp, 1979*; *Gashout & Ormond, 1979*; *Monteiro,*

*Russo & Solé-Cava, 1998*). The rationale behind the use of molecular markers to study asexual reproduction is that it is extremely unlikely that two sexually produced individuals will be identical over a large number of polymorphic loci. Putative clone mates are, thus, those individuals in the population analyzed that have identical multiloci genotypes when the cumulative probability of that identity is very small (*Monteiro, Russo & Solé-Cava, 1998*). Using amplified fragment-length polymorphism (AFLP) markers (*Vos et al., 1995*), a well-established method for cnidarian genotyping (*Amar et al., 2008*; *Douek, Amar & Rinkevich, 2011*; *Brazeau, Lesser & Slattery, 2013*), we demonstrate that the *Alicia sp.* population held by *L. leptochelis* has a particularly small number of genotypes. Remarkably, each pair of sea anemones held by a single crab is identical, strongly suggesting that they are clones obtained by splitting a single sea anemone into two new ones. This is congruent with our behavioral observations of theft and splitting among the crabs, indicating that crab induced splitting is a major reproductive strategy of the sea anemone. Furthermore, the significant size correlation between sea anemone pairs from wild caught crabs adds credence to this assertion. It is still unclear how, where and when the crab obtains its sea anemones in nature. It is reasonable to assume that although splitting and theft occurs in nature, it does not exhibit the full picture of the acquisition mechanism. The AFLP profiles of 6 out of 8 sea anemone pairs were identical, containing representative pairs from both locations sampled. Of the remaining two genotypes, one originates from Tur-Yam and the other from Red-Rock. These beaches are approximately 3 km apart, separated by a large man made barrier in between them, the Eilat port complex, spanning approximately 850 m.

The remaining two genotypes are from each location (Fig. 6 and Table 6). Due to strict collection regulations and a general scarcity of the animals, we limited the genetic part of the study to a small sample size. Thus, the scarcity of the less frequent genotypes at each of the two sample localities may be due to a sampling bias, not fully reflecting the sea anemones population level genetic profile. *Brazeau, Lesser & Slattery (2013)* found that even when using a small sample size, in a limited geographic area, AFLP is a powerful tool for investigating genetic differences among individuals and warrants strong reconsideration as a tool in population genomic analysis, particularly when sampling is constrained. Another crab induced behavior which presumably contributes to the maintenance of a crab specific sea anemone genotype is molting. Over 20 times (Y Schnytzer, 2008, unpublished data) throughout the course of this study we observed molting. Typically, a newly molted crab was found in its aquaria with sea anemones in its claws while the exuviae was deprived of them. Upon the completion of molting, the crab would retake its sea anemones from the claws of the exuviae, each to its original claw (Video S4).

## CONCLUSIONS

We have shown that the *Lybia*-sea anemone acquisition mechanism is composed of a unique behavioral repertoire. Both sea anemone theft and splitting are highly significant behaviors in laboratory held *L. leptochelis*. The genetic analysis of sea anemone pairs from wild caught crabs show genetic identity within the pairs and also between pairs, this

provides further support to the hypothesis that the genetic profile of the sea anemone population are modulated to some extent by the crab behavior. This association is a rare and perhaps unique example of one animal which not only regulates the feeding and growth of its associate (*Schnytzer et al., 2013*), but also controls its asexual reproduction. The exploration of the genetic profiles of the so far not found freely living *Alicia* sp. as well as expanding the study to further *Lybia* populations would greatly enhance our understanding of the role played by the crabs through splitting and theft in affecting the genetic diversity of their cnidarian associates.

## ACKNOWLEDGEMENTS

We thank the staff at the Interuniversity Institute for Marine Sciences in Eilat for their hospitality and assistance with the field work. We thank DG Fautin, AL Crowther and D Guinot for identifying the animals. We are grateful to Adi Schnytzer, Jennifer Benichou Cohen Israel and Dr. Yury Kaminer for helping with the statistical analysis, and E Costi for helping to set up the lighting system and for technical support with the aquarium room. A special thanks to RivQua Bar-Noy for drawing the illustrations. This research is part of the MSc thesis requirements for Yisrael Schnytzer and Yaniv Giman under the supervision of Yair Achituv and Ilan Karplus at The Mina and Everard Goodman Faculty of Life Sciences, Bar Ilan University, Israel.

### Funding

The authors received no funding for this work.

### Competing Interests

The authors declare there are no competing interests.

### Author Contributions

- Yisrael Schnytzer and Yaniv Giman conceived and designed the experiments, performed the experiments, analyzed the data, wrote the paper, prepared figures and/or tables, reviewed drafts of the paper.
- Ilan Karplus and Yair Achituv conceived and designed the experiments, contributed reagents/materials/analysis tools, wrote the paper, reviewed drafts of the paper.

### Field Study Permissions

The following information was supplied relating to field study approvals (i.e., approving body and any reference numbers):

The animals were collected and maintained within the guidelines of the Israel Nature and National Parks Authority (Permit no. 26103/2006/7/13).

### Data Availability

The tables within the manuscript contain all the relevant raw count data.

## Supplemental Information

Supplemental information for this article can be found online at http://dx.doi.org/10.7717/peerj.2954#supplemental-information.

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
