# Peer review of "Boxer crabs induce asexual reproduction of their associated sea anemones by splitting and intraspecific theft"

_PeerJ, doi:10.7717/peerj.2954_

## Round 0.1 · original submission · Major Revisions

Please review and address the comments from both reviewers, as per the attached.

Reviewer 1 ·

Basic reporting

The manuscript in general needs to be heavily edited with regards to grammar; sentence structure and the repetitive nature of sentences particularly. I found myself rereading many parts in an attempt to find the real meaning. This unfortunately distracted me from being able to review the strength and validity of the experimental data as I was constantly double checking the meaning. I have gone through the manuscript and added some examples where ambiguity arose.

The introduction is lacking comment on reproduction strategies/studies of the Alicia genus itself.

Reconsideration of the use of a Video, a figure and the amount of text explaining Figure 5. It is repetitive presentation of results.

Videos could be a shorter duration - especially where nothing is happening (eg Supp video 2) with a line of explanation in text that it is not the entire video.

Experimental design

Research questions were clearly defined at the outset.
The experimental design appears to have followed an ethical protocol
There seems to be a biased towards sampling more male crabs - it is unclear why you would not sample the same number of each sex.
The same applies for why you would hold more trials for right hand vs left hand sea anemones and the number of day vs night trials.
It is also unclear if the same crabs were used in the splitting and the theft trials - or was it a different cohort? this could introduce a bias if the same cohort were used.
Why were sea anemones from the splitting and theft behaviour studies not included in the AFLP portion of the genetic experiment design. That is prior to the theft and splitting, to show genetic diversity and then applying the findings to the wild population?
I am unclear as to the point of statistical aspect of the study and how that was relevant to any of the proposed research questions. That is more related to a study of phenotypes than genotype diversity. Perhaps you need to change/add focus of the paper.

Validity of the findings

From the title I was expecting a paper on genetic diversity. However, it was more a paper on the process and mechanisms of the sea anemone theft and subsequent regeneration of sea anemones.

I found that I had to keep going back to the questions to find the relevance from the discussion. The discussion does not adequately address the questions posed.

The amount of unpublished data referenced in the discussion is concerning - if it is pertinent to the paper at hand for discussion it should be presented in the results.

I find that concluding that Alicia sp. reproduction is controlled solely by the crab is a large assumption. Certainly preliminary findings indicate it, but you cannot be certain unless you had an experiment whereby you studied Alicia sp. separated from the crabs. Some sea anemones have the ability to change their reproduction methods as required. As so little work has been done on this particular species, and on many species, the documented reproduction strategy at one point in time is valuable data.

Additional comments

I have attached annotated PDF with examples of where changes should reviewed. In some cases I have only made a note on the first instance of an occurrence. The assumption would be that the manuscript is reviewed for all instances following the first occurrence. The results presented and their relevance needs to be re-evaluated.

Annotated reviews are not available for download in order to protect the identity of reviewers who chose to remain anonymous.

·

Basic reporting

I have found the paper extremely interesting, well written, with sound results and a good discussion. In my view this manuscript adds useful information to this thematic.

Experimental design

No comments.

Validity of the findings

The size of the sample used to AFLP analysis is small (16 anemones from 8 crabs). This fact does not affect the results on behaviour but precludes some conclusions on genetic diversity...perhaps the title should be changed or toned down. If sampling size cannot be increased for the genetic analysis (due to restrictions) at least is should be stated as a future interesting and necessary work.

Additional comments

The manuscript “Splitting and theft of sea anemones by boxer crabs and their effect on the genetic diversity of their cnidarian associates” (#12559), written by Schnytzer and colleagues aims to describe the effect of an amazing symbiotic behaviour on the genetic diversity of one of the involved species.

I have found the paper extremely interesting, well written, with sound results and a good discussion. In my view this manuscript adds useful information to this thematic. Nevertheless, I have some concerns and, in my opinion, there is still room to a few improvements.

My major concerns are the following:

Abstract - In my opinion it should be rewritten, it has too many results. The crab behaviour is remarkably interesting but perhaps should not be thoroughly described in the abstract. From the other hand the references on symbiosis were all left out...

The size of the sample used to AFLP analysis is small (16 anemones from 8 crabs). This fact does not affect the results on behaviour but precludes some conclusions on genetic diversity...perhaps the title should be changed or toned down. If sampling size cannot be increased for the genetic analysis (due to restrictions) at least is should be stated as a future interesting and necessary work.

Smaller concerns:

Introduction: It would be nice that the last sentence was used to summarize the major aims of this interesting study.

Suggestions:

It would be extremely interesting to understand how is this sea anemone phylogenetically related with the remaining species of its genus, using adequate markers. In my opinion this new species should be described because, due to its exclusively symbiotic existence, it will probably deserve a conservation status.

I hope my suggestions can help the authors to improve the manuscript.
Joana Robalo ([email protected])

---

## Round 0.2 · accepted · Accept

Thank you for addressing the reviewers comments in detail. This has improved your already very interesting manuscript.

Reviewer 1 ·

Basic reporting

This manuscript now reads clearly and is excellent - meets all criteria

One small grammatical change suggestion to the sentence

"Crabs of both genders with and without sea anemones were equally likely to initiate a fight (binomial test, P = 0.6291, N = 17)"

should read

Crabs of both genders, with or without sea anemones, were equally likely to initiate a fight (binomial test, P = 0.6291, N = 17).

Experimental design

Meets all criteria

Validity of the findings

Meets all criteria

Additional comments

The paper reads exceptionally well now and is very clear. The title change was important and reflects your body of experiments. I appreciate the effort to address all the comments and I think it has dramatically increased the quality of your paper. Your work is an important contribution to reproduction studies of sea anemones.

·

Basic reporting

The manuscript “Splitting and theft of sea anemones by boxer crabs and their effect on the genetic diversity of their cnidarian associates” (#12559), written by Schnytzer and colleagues aims to describe the effect of an amazing symbiotic behaviour on the genetic diversity of one of the involved species. It is now presented in a reviewed form, integrating most suggestions made by two referees.

Experimental design

I think suggestions were addressed accordingly and methodological issues were clarified.

Validity of the findings

In my opinion suggestions were addressed accordingly.

Additional comments

The manuscript “Splitting and theft of sea anemones by boxer crabs and their effect on the genetic diversity of their cnidarian associates” (#12559), written by Schnytzer and colleagues aims to describe the effect of an amazing symbiotic behaviour on the genetic diversity of one of the involved species. It is now presented in a reviewed form, integrating most suggestions made by two referees.

Specifically considering my suggestions I think they were addressed accordingly.

Joana Robalo
([email protected])